# Experimental Studies of Concrete-Filled Composite Tubes under Axial Short- and Long-Term Loads

**DOI:** 10.3390/ma13092080

**Published:** 2020-05-01

**Authors:** Marcin Abramski, Piotr Korzeniowski, Krzysztof Klempka

**Affiliations:** 1Faculty of Civil and Environmental Engineering, Gdansk University of Technology, 80-233 Gdansk, Poland; 2Institute of Technology, The State University of Applied Sciences in Elbląg, 82-300 Elbląg, Poland; p.korzeniowski@pwsz.elblag.pl; 3Faculty of Geoengineering, University of Warmia and Mazury in Olsztyn, 10-724 Olsztyn, Poland; kik@uwm.edu.pl

**Keywords:** fiber reinforced polymer, concrete-filled FRP tube, column, experiment, long-term loading

## Abstract

The paper presents experimental studies on axially compressed columns made of concrete-filled glass fiber reinforced polymer (GFRP) tubes. The infill concrete was C30/37 according to Eurocode 2. The investigated composite pipes were characterized by different angles of fiber winding in relation to the longitudinal axis of the element: 20, 55 and 85 degrees. Columns of two lengths, 0.4 m and 2.0 m, were studied. The internal diameter and wall thickness of all the pipes were identical and amounted to 200 mm and 6 mm, respectively. The mean values of two mechanical properties, elasticity modulus and compression strength, were determined. These properties were determined for longitudinal compression and for circumferential tension. The graphs of longitudinal and peripheral deformations of polymer shells as a function of load level are presented both for empty tubes and for concrete-filled ones. The results of long-term investigations of three identically made 0.4 m high concrete-filled GFRP tubes are also presented.

## 1. Introduction

### 1.1. Concrete-Filled Fiber Reinforced Polymer (FRP) Tubes (CFFT) Column Concept

CFFT columns are a modification of CFST (concrete-filled steel tubes) columns commonly used in construction [1,2]. They differ only in the type of material used as reinforcement. The CFST columns consist of a steel casing filled with concrete on the inside. In CFFT columns, a composite pipe is used instead of a steel pipe. The abbreviation CFFT has been adopted in the literature on the subject [3,4,5,6,7]. The interest in the use of non-metallic reinforcement results from the desire to increase structure durability, and there has been a significantly increased interest in FRP (fiber reinforced polymer) composite materials. Construction is a branch of industry into which composites came relatively late; before that they were used in the automotive industry, aviation industry, sailing, sports equipment production and others. Many years ago such countries as the USA, Canada and Japan implemented guidelines for designing concrete structures reinforced with composites (e.g., [8]). Some European countries are in the process of implementing such design guidelines and others have done so over the past several decades in order to reinforce concrete structures with composite tapes and fabrics. A number of such applications can also be seen in Poland. In recent years the technology of concrete structure reinforcement using composite prestressing lines has been successfully implemented.

The use of FRP (fiber reinforced polymer) pipes for making columns is an interesting field of application of composites in concrete structures. Originally, the application of composites in concrete columns involved reinforcing them by sticking composite fabric bands in the overload zones. This well-known and widely used technology [9,10] has two main advantages. The first is the reinforcement of the column’s concrete, which, when subjected to axial compression with limited freedom of lateral deformation, increases its strength and makes it able to carry heavier loads. The second advantage is that it increases the resistance of the column in the post-critical area. In an emergency, the circumferentially reinforced column does not crumble, but, even when it reaches large deformations and exceeds its capacity, it is able to carry significant loads. This mild destruction mechanism is important in earthquakes and terrorist attacks, as the failure of one column does not immediately transfer all loads from a heavily deformed column to adjacent columns. In many cases, this protects the construction structures from a so-called progressive catastrophe, which is well-recognized and proven in the case of spiral reinforced concrete columns and CFST columns. Current research is aimed at testing whether there are similar advantages, in addition to the undisputed corrosion resistance, of shells made of polymer pipes reinforced with glass fiber, basalt or carbon fiber [11,12].

One of the main disadvantages of steel is that it is subject to corrosion, and therefore it is necessary to incur additional costs related to maintaining load-bearing structures in good technical condition during the operation of the structure. In case of structures using corrosion-resistant composite reinforcement, significant maintenance costs are eliminated. Additionally, the concrete inside CFFT columns is fully protected against environmental impacts. Unfortunately, each material has its own disadvantages, and the main problem with composites is their low fire resistance.

### 1.2. Examples of Applications of CFFT Columns in Engineering Construction

CFFT columns were used for a road bridge along road no. 40 over the Nottoway River in Virginia, USA [13,14]. One of the supports is built from a row of CFFT piles, which protrude over the ground surface at a height of about two meters and are topped with a reinforced concrete cap. The piles are 13.1 m long, 62.5 cm in external diameter, and the wall thickness of the glass fiber reinforced polymer pipe is 5.65 mm.

A more typical area of application for CFFT piles is the shielding structures of sea piers, in which traditionally used reinforced concrete or CFST piles are subject to accelerated corrosion due to the salinity of the water. The corrosion is particularly evident in the zone of the variable water table. One of many examples is the construction of a munitions dock pier in the military port of Hadlock on Indian Island in Washington, USA [15]. The piles are 28.3 m long, have an external diameter of 42 cm, and the wall thickness of the glass fiber reinforced polymer pipe is 7 mm. Other similar applications of CFFT piles for marine engineering can be found, e.g., in [16].

## 2. Materials and Methods of Experimental Studies on CFFT Columns

The authors carried out research on six concrete-filled poles made of composite polymer pipes reinforced with glass fiber, produced by the Polish company Vitreo. Half of the pipes were 400 mm long and the other half were 2000 mm long. The internal diameter of the ordered pipes was 200 mm. The wall thickness of the pipes was 6 mm. The pipes were characterized by three different angles 20°, 55° and 85° of glass fiber cross winding in relation to the longitudinal axis of the pipe.

The list of all the tested elements is presented in Table 1. All the elements were axially compressed. The support was double-jointed. Shaft joints were used (Figure 1). As a result of the above-mentioned solution, the buckling length of each element increased by 8 cm (double the thickness of steel sheet—40 mm). Using such a support imposed a buckling plane perpendicular to the axis of the shaft. Steel clamps were used at the top and bottom ends of the column to prevent damage to the tested elements in the zones of the column support (Figure 1).

Six strain gauges were placed on the outside of the polymer pipes halfway along their length—three in a vertical position and three in a horizontal position, at equal spacing of one third of the circumference of the element (Figure 2).

Unlike other researchers [3,4,5,7], the authors subjected the applied composite materials of the pipes to extensive experimental studies in order to identify their mechanical parameters, both in the longitudinal and peripheral direction.

### 2.1. Composite Pipes

#### 2.1.1. Geometric and Technological Parameters of Pipes

The glass fibers used for the production of the pipes were in the form of continuous rovings with a linear density of 2400 g/m (2400 tex). They were produced from E-type glass fibers in Slovakia by the American company Johns Manville. Polyester resin, with the trade name Crystic 2-420 PA, was produced in Croatia by the British company Scott Bader. The manufacturer declared the tensile strength of the hardened resin between 44 and 46 MPa, and the tensile modulus between 3.03 and 3.68 GPa. The resin hardener, called Butanox M-50, was produced by the Dutch company AkzoNobel. The pipes’ internal diameter declared by the manufacturer was 200 mm, and the wall thickness was to be between 5.5 and 6.0 mm. The former corresponded exactly to reality, while the wall thickness measured by the authors ranged from 4.9 to 8.8 mm. The measurement was made on ring samples cut from pipes (Figure 3). The average values of the measured pipe wall thicknesses are presented in Table 2.

The glass fiber content of the composite, as stated by the manufacturer, was 60%. This content was verified on the basis of the standard method [17] used in the USA, which consists of firing a polymer resin from a composite sample at 535 °C and weighing the remaining part of the sample, i.e., the glass fibers themselves. The fibers’ content in weight terms for pipes with winding angles of 20°, 55° and 85° was 69.4%, 58.4% and 72.4%, respectively. The fiber winding angles measured by the authors were in accordance with the values declared by the manufacturer. Figure 4 shows, for example, a photograph of a column fragment made of a pipe with a winding angle of 20 mm, taken after the completion of the load-bearing capacity test. The photograph shows lines that intersect with the column axis at an angle of 20°, done with AutoCad.

#### 2.1.2. Longitudinal Compression Strength of Pipe Composite

The mechanical properties, compressive strength and modulus of elasticity of the pipe material were determined through the testing of 400-mm-long pipe sections (Figure 1). Prior to the testing, the pipe ends were secured from the inside with 100-mm-thick concrete plugs and equipped with 60-mm-wide and 8-mm-thick steel clamps from the outside to avoid crushing of the pipe in the compression force transfer zones from the testing machine to the pipe.

Experimental research of the pipes’ composite strength to longitudinal compression *f*_FRP,c_ was carried out with the use of a hydraulic press produced by the Swiss company Walter and Bai AG, model 102/5000-HK4, with a load capacity of 5000 kN and piston extension from 0 to 100 mm. The load was controlled by the displacement, i.e., by the constant extension of the press piston. The press tooling made it possible to digitally record the measurement results. The measurement of the piston advancement and force values was carried out discretely at a time interval of several seconds. The duration of a single experiment was several dozen minutes.

The mode of destroying the pipes is shown in Figure 5. In each case, a clear resin crack was formed in the composite pipe along the fibers, consistent with the direction of their winding. The cracks always appeared suddenly, unannounced. After the crack had occurred, it was possible to further increase the displacement of the press piston, as the pipe did not completely lose its load-bearing capacity. When the vertical displacement increased further, further cracks appeared, but the load-bearing capacity of the pipe was determined by the first crack. The behavior of the pipes in increasing vertical displacement forced by the movement of the press piston is discussed and shown further in Section 2.1.3.

The highest load capacity in the group of empty short pipes was obtained by a sample with fiber winding orientation closest to the longitudinal axis of the pipe. The load-bearing capacity of a pipe with a 20° winding angle had about 100% advantage over the other two tubes. It was probably obtained thanks to the compression strength of the glass fibers themselves. For pipes with fiber orientation more deviated from the vertical, the decisive factor was the compression strength of the resin itself, many times lower than the strength of reinforcing fibers (values between 120 and 180 MPa and about 800 MPa, respectively, according to [18]). The dependence of composite strength in axial compression on the angle of glass fiber winding is presented in Table 2.

#### 2.1.3. Elasticity Modulus and Ultimate Strains of Pipe Composite in Longitudinal Compression

The elasticity modulus was measured during the load capacity tests of 400 mm sections of empty pipes, described in Section 2.1.2. Three vertically and three peripherally directed strain gauges were applied at half of the height of all samples in three groups of two strain gauges (Figure 6). According to the strain gauges manufacturer (Vishay Company, Malvern, PA, USA) the engineering tolerance of the gauge factors was 0.5%.

These three groups were installed at a mutual distance of 120° around the circumference of the pipe. One strain gauge measuring the longitudinal deformations of the pipe and one strain gauge measuring the circumferential deformations were glued side by side. The strains of pipes were measured by the gauges discretely, every 0.5 mm of the press piston advancement. In this way, nine readings were obtained for each of the strain gauges in the case of the pipe with the winding angle θ = 20°, sixteen readings for θ = 55° and twenty readings for θ = 85°. The graphs presented in Figure 7 were obtained by averaging readings from three vertically installed strain gauges (solid lines) and three horizontally installed ones (dotted lines). The strain developed in the whole range of tests until the destruction was approximately linear. Based on the obtained results, the elasticity modulus of composites in longitudinal compression was determined. For each of the samples, almost the entire *σ*(*ε*) graph was taken into account, omitting only the falling part of the curve. It was assumed that the approximation line had to start at the beginning of the coordinate system. The resulting elasticity modulus values of three types of FRP pipes are given in Table 3. The table also gives the values of composite ultimate shortening when maximum compressive stresses are reached. These values were calculated according to Hooke’s law using the compressive strength values of the composite given in Table 2.

The samples with the smallest winding angle in relation to the longitudinal axis (angle θ = 20°) reached the highest value of elasticity modulus in the longitudinal direction. Both remaining samples (winding angles θ = 55° and θ = 85°) were characterized by several-times-lower elasticity modulus, while the values of elasticity modulus of these samples were similar. In this respect, the results turned out to be similar to the results of composite longitudinal compression strength *f*_FRP,c_ obtained in the tests described in the previous subchapter. It is worth noting that the value of elasticity modulus for glass fibers is about 75 GPa according to [18], for polyester resin they are between 2.8 and 3.5 GPa according to [18] and between 3.03 and 3.68 GPa according to the manufacturer’s specification, i.e., Scott Bader.

#### 2.1.4. Circumferential Tensile Strength of Composite Tubes and Modulus of Elasticity at Circumferential Stress

The tests of *f*_FRP,circ_ circumferential strength of the pipe and *E*_FRP,circ_ elasticity modulus were performed with the standard method which consisted in tearing the ring cut out from the pipe. This method, called the split disk method, is used in Polish [19], European and American [20] standardization. For each group of pipes, five rings cut from the pipe were prepared. The width of the rings was 25 ± 2 mm. Figure 3 and Figure 8 show photographs of sample specimens. In accordance with the recommendations of standard [19], the rings were weakened before the test through milling a notch on both sides in one place at their circumference (see Figure 8), where the width of the ring was reduced from 25 mm to 15 mm. Strain gauges were glued onto the notches (see Figure 8) for testing the peripheral modulus of elasticity of a composite. The rings were tested using a fixture for the testing machine, which is shown in Figure 9 together with the ring sample after the test.

According to the strain gauges manufacturer, the engineering tolerance of their gauge factors is 0.5%. A measuring uncertainty test was conducted for strains taking into account the tolerance of strain gauge factors along with other uncertainty factors: change in gauge factor and in amplifier characteristics due to change in ambient temperature, tolerance of measuring equipment including electric cables, imprecise gauge installing and a degradation in the measuring equipment sensitivity since the last calibration. The resulting total measuring uncertainty was equal to 1.4%.

Having the relationship between the strength measured by the testing machine and the composite strain at the notch point of the ring, as well as the pipe wall width and thickness measured at the notch before the test, the relationship of *σ*-*ε* was determined for each of the fifteen tested rings (Figure 10). A characteristic feature of the *σ*-*ε* graphs for each of the composites tested is negative strain (shortening) in the initial phase of the experiment. This was induced by local bending of the ring at the point where two semi-disks meet. In this place, the ring was locally straightened, which caused compression of its outer fibers (where the strain gauges were installed) and tension in the internal fibers. After reaching a certain level of tensile force, the local bending effect passes and the further development of the σ-ε relationship is almost linear.

In order to determine the modulus of elasticity at circumferential tension, the experimentally obtained *σ*-*ε* graphs were considered only within their parts of tensile strains. The parts of compressive strains of the outer ring fibers were completely removed from the charts (Figure 11) in order to omit the local bending effect on the calculation. Such an effect does not occur in CFFT columns. The modulus of elasticity *E*_FRP,circ_ was calculated using the least squares method. For this purpose, Microsoft Excel software was used, which enabled an automatic identification of the linear function.

The results of circumferential strength of the composite and modulus of elasticity at circumferential tension are presented in Table 4. The table also shows the values of the ultimate circumferential elongation of the composite *ε*_FRP,u,circ_. These values were calculated using Hooke’s law.

### 2.2. Concrete Filling Composite Tubes

All CFFT (concrete-filled fiber reinforced polymer tubes) columns were made simultaneously from a single shipment of concrete. The ordered concrete was class C30/37 according to Eurocode 2 [21], with a w/c ratio equal to 0.52, S3 consistency, according to standard [22], maximum aggregate grain size 16 mm and exposure classes according to standards [23]: XC3, XD2, XF1, XA1. For the production of the concrete, regular hardening cement CEM 42.5 N was used. The admixture in the form of superplasticizer was used and fly ash was used as a filler.

Concrete compression strength and modulus of elasticity in compression tests were carried out on standard concrete cylinders with a diameter of 150 mm and a height of 300 mm. Concrete samples were stored from the moment they were made to be tested in the same thermal-humidity conditions as the columns, i.e., at a temperature of about 20 °C, with relative air humidity of about 50%. Both tests of concrete samples were carried out with the use of a German Heckert strength press, model DP1600, with an operating range from 0 to 1600 kN. Elasticity modulus of concrete was tested using a compressometer-extensometer according to standard [24].

Concrete used for the construction of axially compressed columns was tested in parallel with the experimental tests of the load capacity of these columns. They were carried out after a period of eighty to ninety-two days from the moment of filling them with concrete. Because almost three months passed between preparing the columns and testing them, it was decided to assign the same strength characteristics of concrete to all six columns and to disregard small differences resulting from the concrete’s varied age. The strength of concrete obtained this way in the uniaxial compression state was 38.1 MPa after 28 d and 42.1 MPa after 86 d. Elasticity modulus of concrete was 31.9 GPa and 33.0 GPa respectively. Concrete density was not investigated.

## 3. Results of the Experimental Studies on Concrete-Filled Tubular Columns

### 3.1. Short Columns

Short columns (items from one to three in Table 1) were destroyed by cracking of the composite jacket as a result of radial deformation (elongations) of concrete. In the shell made of polymer pipe reinforced with glass fibers, which limits the freedom of these elongations, tensile stresses were created which led to the destruction of the shell. By limiting the lateral deformations of concrete, the composite jacket exerted radial stresses on the concrete, causing a triaxial state of compressive stress in the concrete core. This increased the compression strength of the core concrete in the direction of the longitudinal compression force. The ability of the shell to limit the lateral deformations of concrete, and thus to generate an increase in lateral stresses on concrete, was greater if the circumferential elasticity modulus of the pipe composite was higher. The best mechanical parameters in this respect were shown by the shell with the fiber winding angle θ = 85° (see Table 4). For a pipe made of this composite, not only the highest column load capacity was obtained (see Table 5) but also its high deformability (Figure 12).

The lowest load capacity was obtained for a column with a fiber winding directed at 20° to the longitudinal axis of the column. This column stood out from the others with the smallest shortening accompanying the maximum destructive force. This reduction was over five times smaller than in the case of the other two columns.

This strong advantage of columns with 55° and 85° winding angles (especially the latter) resulted from a more effective reduction in the column deformation in the circumferential direction. The shells of these columns protected the concrete core from damage, probably even after the limit elongations were exceeded. This was clearly demonstrated by the fact that the samples were barrel-shaped after destruction.

While discussing the “force—vertical displacement of the piston” graphs (Figure 12), it is important to point out their bilinear character in the case of two stronger columns with the fiber winding angle of 55° and 85°. This does not apply to the first phase of column operation, to the level of about 100 kN, which may be explained by the gradual adjustment of the bearings’ joints to the increasing compressive force. When the force of 1500 kN to 2000 kN is reached, respectively for columns with fiber winding of 55° and 85°, the columns become noticeably ductile. This is caused by the softening of the concrete in the core, whose transverse deformations are prevented by the glass fiber reinforced polymeric shell. This results in a triaxial stress state in the concrete. For the weakest column, this phenomenon does not occur at all.

Figure 13 shows the development of average longitudinal and circumferential deformations obtained in the study of the discussed columns. A clear predominance of columns made of tubes with a winding angle of 85° is apparent. For comparison, Figure 7 shows an identical relation for empty composite pipes. It is easy to notice that in this case, the most favorable are pipes reinforced with fibers with the winding angle of 20°, and the least favorable are those with the winding angle of 85°. However, the results presented in Figure 12 are decisive for the selection of the optimum winding angle of the fibers in the composite pipe from the point of view of column load capacity and deformation. As can be seen, the longitudinal mechanical parameters of the composite pipe are of secondary importance in relation to its circumferential parameters.

### 3.2. Long Columns

Long poles (items four to six in Table 1) got destroyed by cracking of the shell along the fibers (the column with 20° winding angle) or by buckling (the other two columns). The appearance of the columns after damage is shown in Figure 14. The most significant test results of these columns are shown in Figure 15 and Table 6.

As can be seen in Figure 15 and Table 6, the group of long columns has not experienced such strong disproportions in load capacity as for the short columns. Nevertheless, the column with the 20° winding angle still had the most brittle destruction mode. However, it was no longer the column with the lowest load-bearing capacity. The higher slenderness of the columns resulted in a lower effect of concrete strength enhancement due to its triaxial stress state. The difference in strength of the shell itself was more significant than in the series of short columns because of the lower load capacity of the long columns. The column made of a pipe with the fiber winding angle of 55° turned out to be the weakest, because on the one hand it was characterized by lower longitudinal strength of the shell and on the other hand by the average ability to create a triaxial compression state in the core’s concrete. In order to confirm the above conclusions, however, a larger number of columns should be examined.

The columns made of fiber-reinforced pipes with the winding angle of 85° displayed the safest post-critical behavior, followed by 55° and 20° angles. This feature can be very important during earthquakes and in emergency situations, preventing the phenomenon of a progressive catastrophe.

Figure 16 shows the development of average deformation of the shell of the tested columns. It is worth noting the significant increase in mean circumferential elongations in the column with the winding angle close to the circumferential one (i.e., 85°), similar to those in the short columns. This increase indicates effective restraining of transverse elongations of concrete also in the long columns, which is surprising.

### 3.3. Long-Term Studies

The test under long-term load was performed for three CFFT columns with a height of h = 400 mm (Figure 17). The outer shell of columns made from FRP pipes was derived from the same batch of material used in the short-term survey of column load-bearing capacity. Three samples were made from the tube with winding angles θ = 20°, θ = 55° and θ = 85°. Concreting of the inside of the pipes was done using the same batch of concrete from which the columns for short-term load capacity testing were concreted.

After concreting, the specimens were placed in an air-conditioned chamber in which the relative humidity was 50% and the temperature 20 °C; they were stored there until the test. An extensometer with a resolution of 1/100 mm and a measuring base of 250 mm was used to read the deformations. Frames were mounted along three lines evenly distributed on the side of the cylinder (every 120°). The tests commenced after sixty-three days, counting from the moment the samples were concreted. The determined average compression strength of concrete after twenty-eight days f_c_(28) = 38.1 MPa was the basis for calculating the level of long-term load in the creep tests. After axial alignment, the specimens were left unloaded in the creep testing machine for one hour. After that time the initial reading was taken and the whole cross-section was loaded. The test was carried out continuously until the stresses σ = 0.46 f_c_(28) were reached (stresses were determined in relation to the cross-sectional area of the concrete core), taking readings at levels 0.36·σ, 0.73·σ and σ. The total load was obtained within ten minutes. The creep tests were carried out with the use of a German Walter Bai strength press, model HKB1000, with the accuracy class 1 and an operating range from 0 to 1000 kN. The measuring uncertainty of the machine indicated in its calibration certificate was in range from 0.25% up to 0.63%, depending on the measured force magnitude.

The first measurement was made after five minutes and the result was considered to be a so-called “temporary deformation”. Then the readings were taken for the period of 392 d. In the first week, readings were made daily, during the next three months, once a week, and after that time, once a month. Based on the obtained results, a curve of rheological deformations was determined taking into account the creep of the concrete and composite FRP as well as concrete shrinkage (Figure 18).

The smallest deformations were obtained for the sample with the 20° winding angle and the largest for the sample with the 55° winding angle. This refers to both immediate and long-term deformations resulting from creep and shrinkage of concrete. It is a consequence of the influence of the angle of glass fiber winding on the deformability of the composite. For pipes with fiber orientation more deviated from the vertical, the deformability of the resin itself is decisive (see Section 2.1.3, Figure 7). It is worth noting that in the study of short CFFT columns, discussed in Section 2.2, at a force of approximately 550 kN, i.e., σ = 0.46 f_c_(28), the largest longitudinal deformations were also observed in samples with a winding angle of 55°. The pipes with winding angle θ = 55° seem to be insufficiently stiff in the longitudinal direction and at the same time insufficiently stiff in the circumferential direction. As a result, the stub CFFT columns made of these pipes deform most severely in the longitudinal direction.

## 4. Conclusions

The conducted experimental research allows the formulation of the following conclusions:columns made of FRP pipes filled with concrete (concrete-filled FRP tubes, (CFFT)) behave in a way that depends on the direction of fiber winding in the pipe;the load-bearing capacity of the CFFT columns is highest when the fibers are wound in a circumferential direction;the difference in load-bearing capacity caused by the direction of the fiber winding can be very large for stub columns and reach more than 150%;for slender columns used in the construction industry, however, the difference will be much lower—in the conducted studies the difference amounted to 33% at the most;brittle failure mode and low post-critical load capacity eliminate columns with a fiber winding angle close to the longitudinal one (in the study in question the winding angle was 20°) from applications in the construction industry;pipes with a winding angle close to the longitudinal one are characterized by increased resistance to longitudinal compression (in the studies described herein by as much as 100%) and higher modulus of elasticity, but these advantages are secondary to the disadvantages described above;in long-term tests, similar to short-term tests, the specimen with a fiber winding angle close to the longitudinal one showed the least strains;the biggest strains in long-term tests were obtained for the specimen with a fiber winding angle of 55°.

## Figures and Tables

**Figure 1 materials-13-02080-f001:**
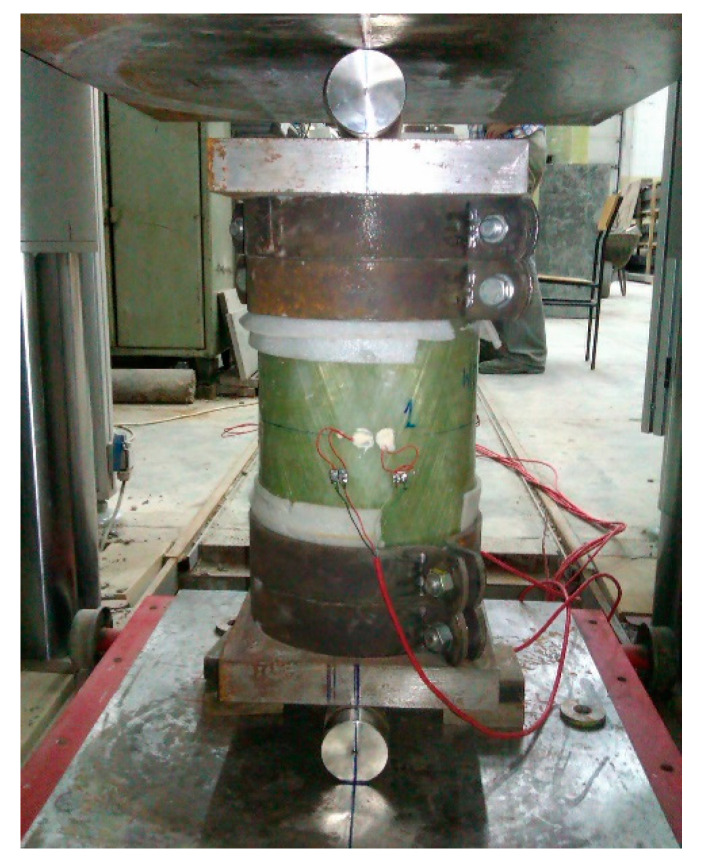
Test element for measuring the mechanical properties of a composite during compression: empty tube with winding angle of 20° on a test stand.

**Figure 2 materials-13-02080-f002:**
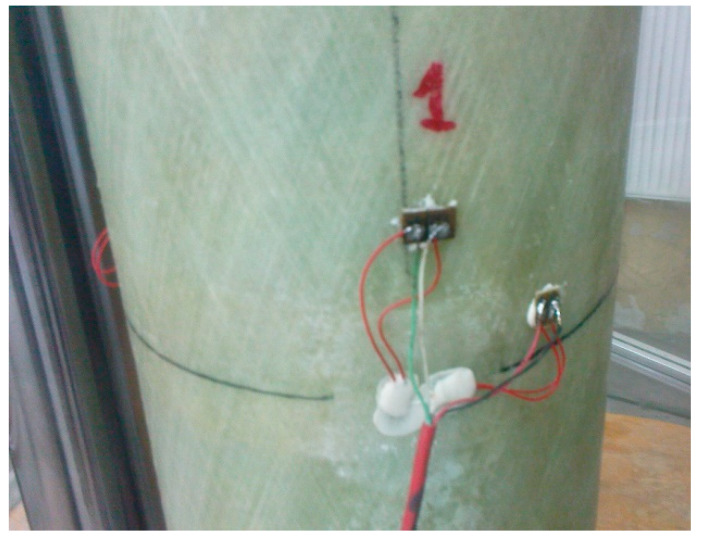
View of a pair of strain gauges at point 1—strain gauges in the picture covered with adhesive to protect against mechanical damage and moisture.

**Figure 3 materials-13-02080-f003:**
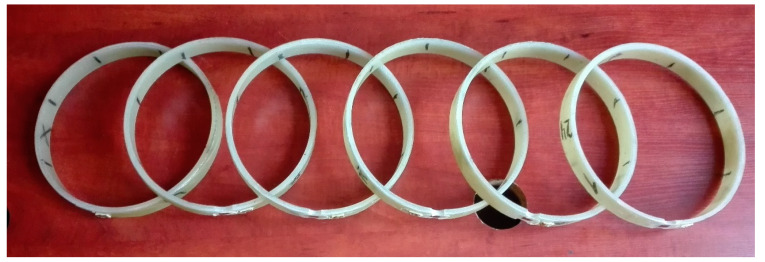
Pipe rims for wall thickness measurements.

**Figure 4 materials-13-02080-f004:**
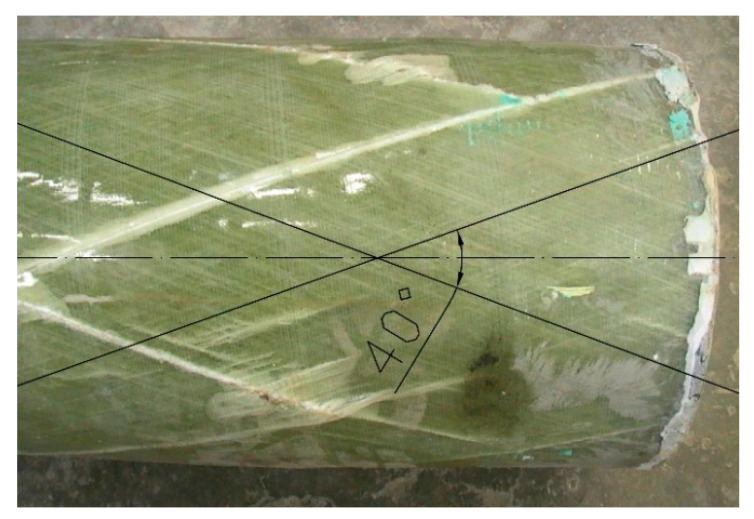
Measurement of fiber winding angle for pipe with declared winding angle 20°.

**Figure 5 materials-13-02080-f005:**
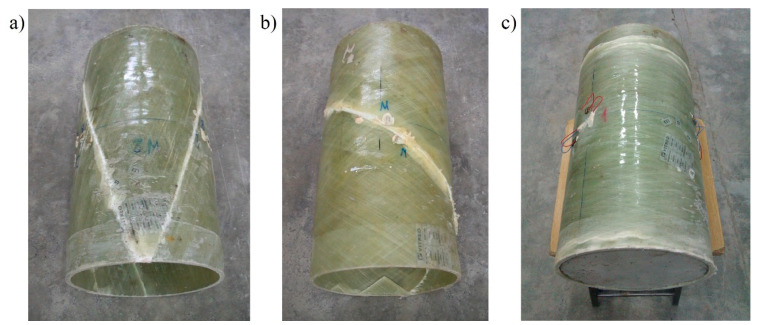
Test samples made of 40-cm-long hollow tube. View after destruction of elements with a fiber winding angle of: (**a**) 20°, (**b**) 55° and (**c**) 85°. Concrete plug visible in the last sample.

**Figure 6 materials-13-02080-f006:**
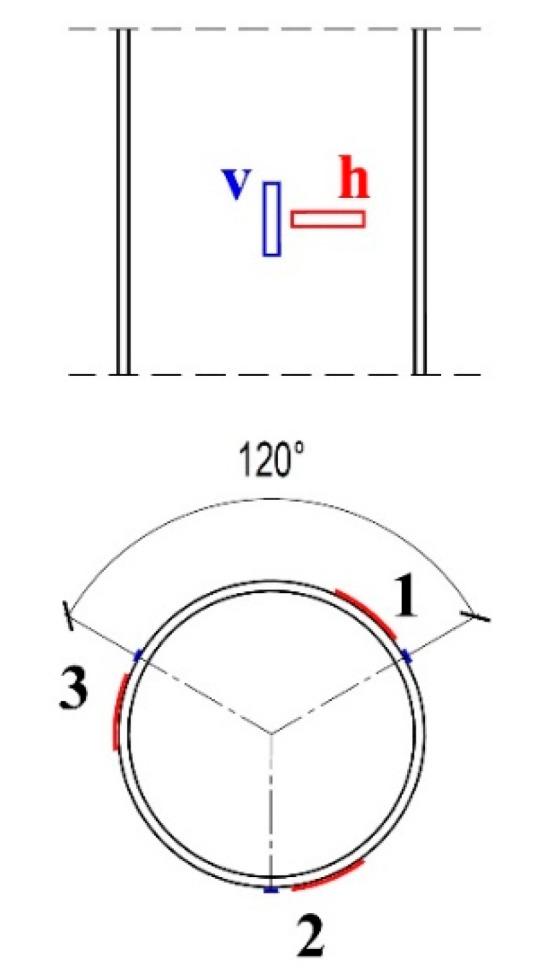
Arrangement of strain gauges to be applied to pipe: v—vertical, h—horizontal.

**Figure 7 materials-13-02080-f007:**
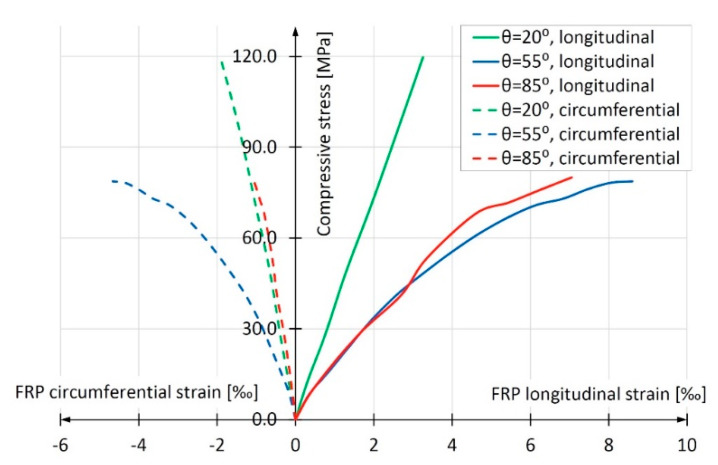
Stress-strain relationship for test elements made of 40-cm-long hollow composite tube.

**Figure 8 materials-13-02080-f008:**
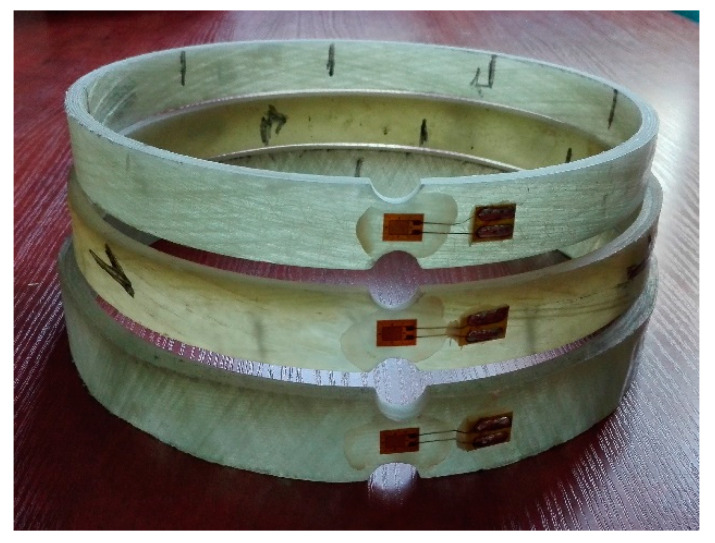
Samples cut from pipes with winding angles of glass fibers θ = 85° (highest), θ = 55° (middle) and θ = 20° (lowest) to test the peripheral tensile strength and modulus of elasticity.

**Figure 9 materials-13-02080-f009:**
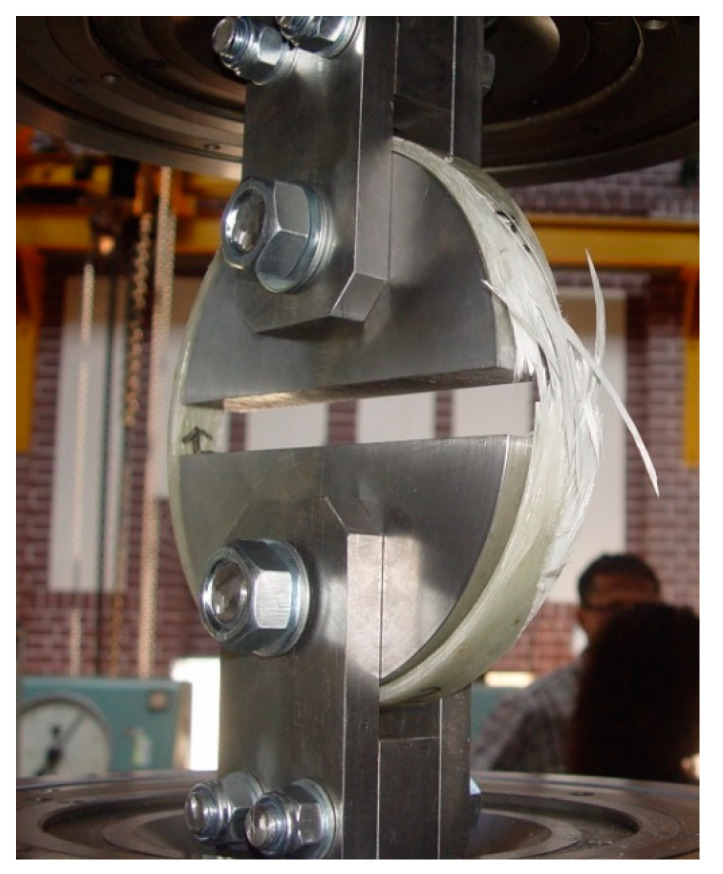
Ring sample of pipe with winding angle θ = 85° on the test stand after completion of experiment.

**Figure 10 materials-13-02080-f010:**
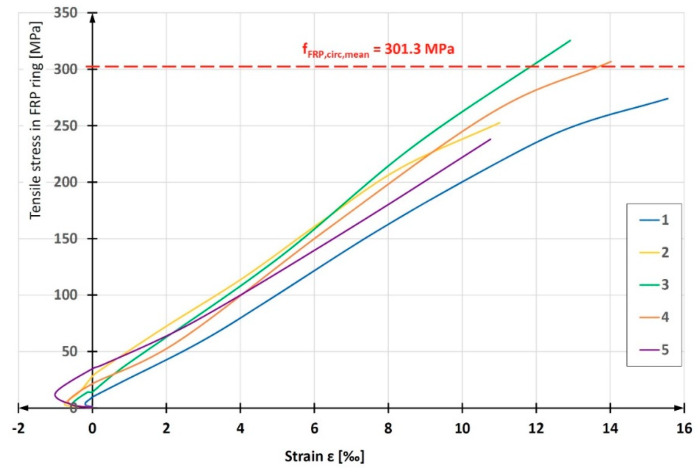
Tensile *σ*-*ε* relationships in circumferential direction, obtained for fiber reinforced polymer (FRP) ring samples of pipe with winding angle θ = 55°.

**Figure 11 materials-13-02080-f011:**
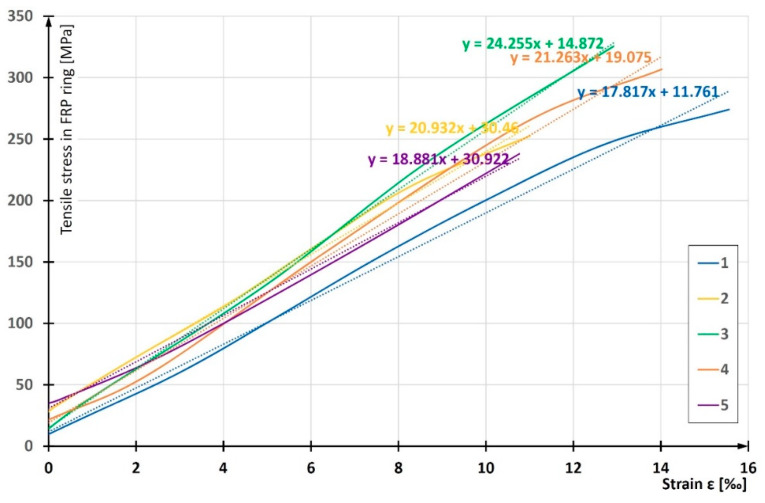
Idea of determining modulus of elasticity at circumferential tension, presented for FRP ring samples of pipe with winding angle θ = 55°.

**Figure 12 materials-13-02080-f012:**
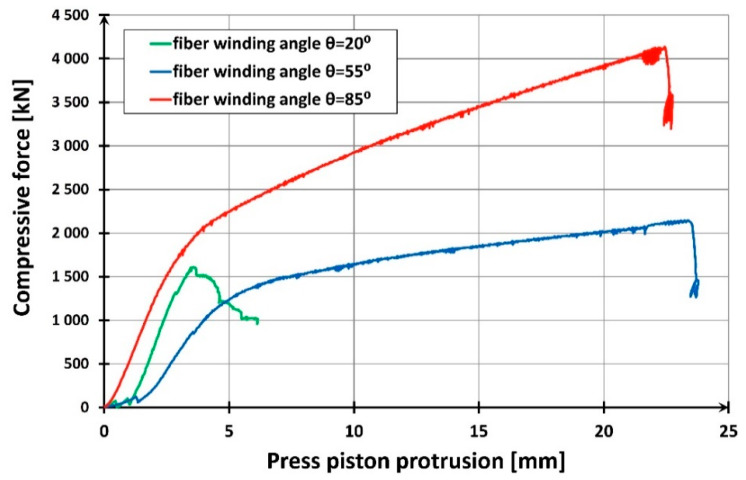
Force-displacement relationship for test specimens made from a 40-cm-long composite pipe filled with concrete.

**Figure 13 materials-13-02080-f013:**
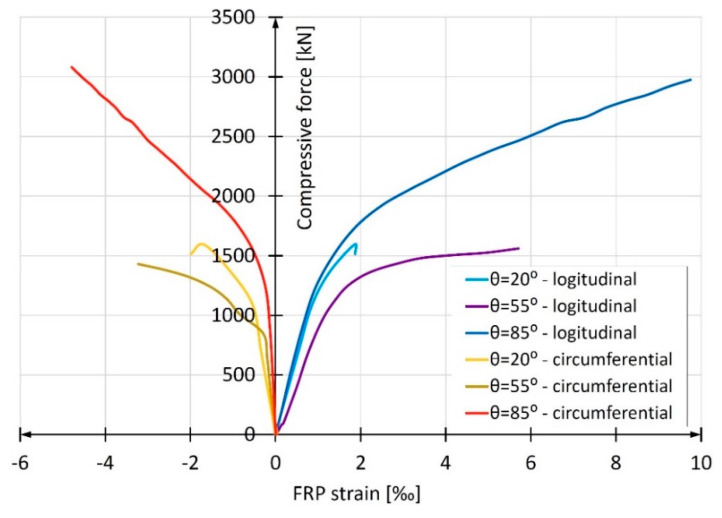
Relation compressive force vs average deformation in short CFFT columns with different fiber winding angles.

**Figure 14 materials-13-02080-f014:**
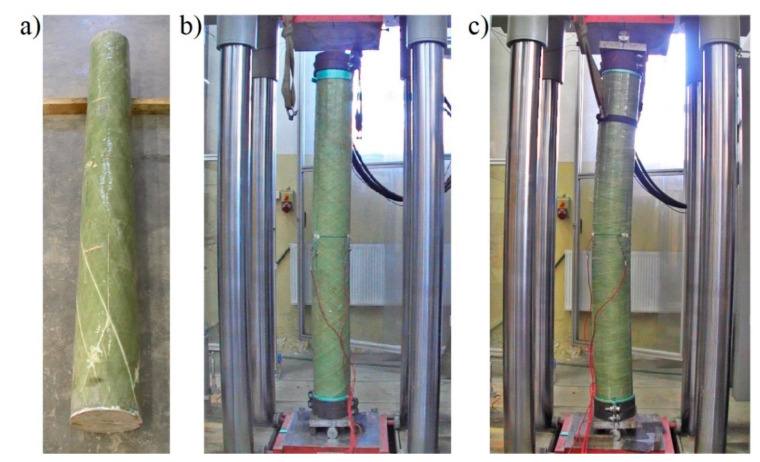
View after failure of 200-cm-long CFFT test specimens with fiber winding: (**a**) 20°, (**b**) 55° and (**c**) 85°.

**Figure 15 materials-13-02080-f015:**
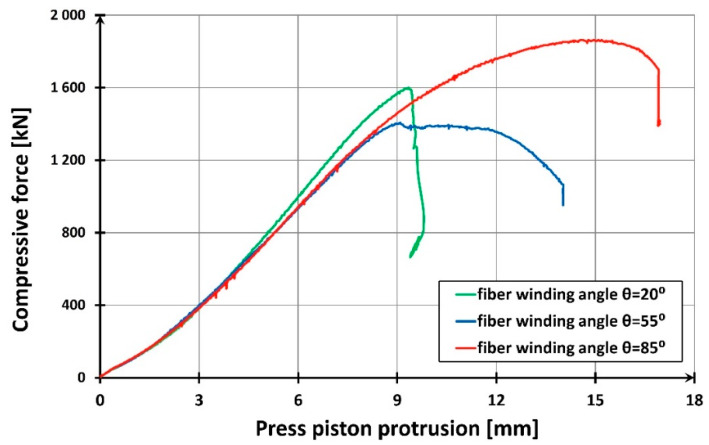
Force-displacement relation for test specimens made of 200-cm-long composite pipe filled with concrete.

**Figure 16 materials-13-02080-f016:**
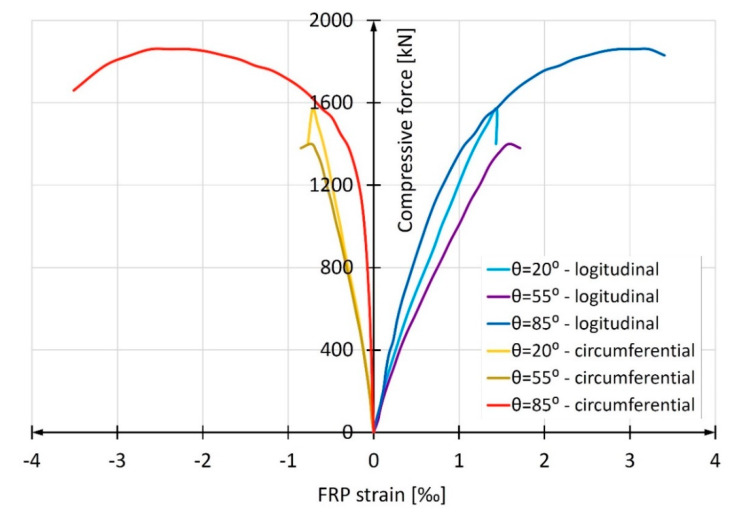
The development of average deformations in the long CFFT columns with different fiber winding angles.

**Figure 17 materials-13-02080-f017:**
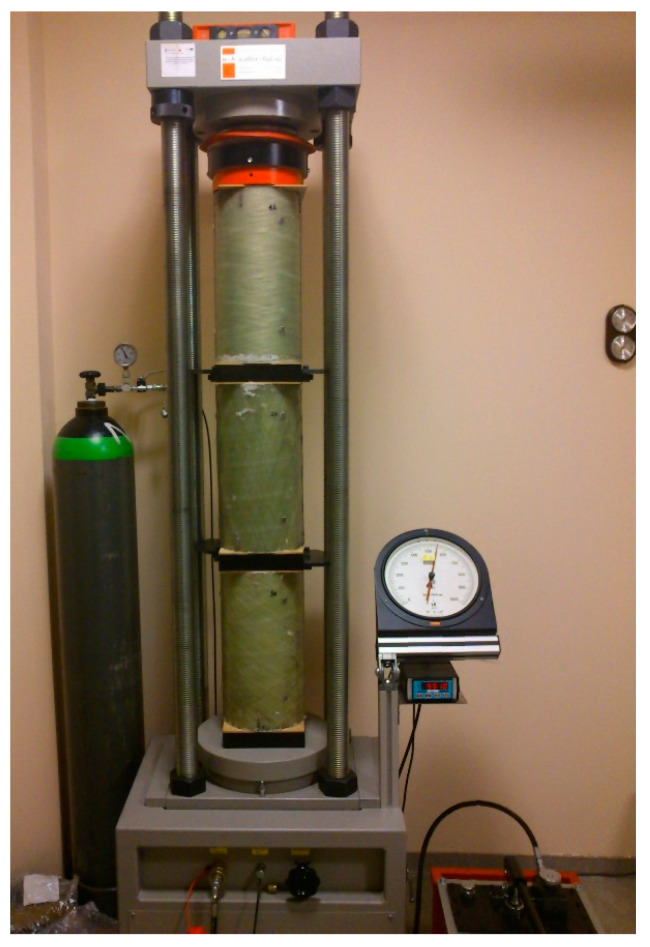
Testing of rheological deformations (creep and shrinkage).

**Figure 18 materials-13-02080-f018:**
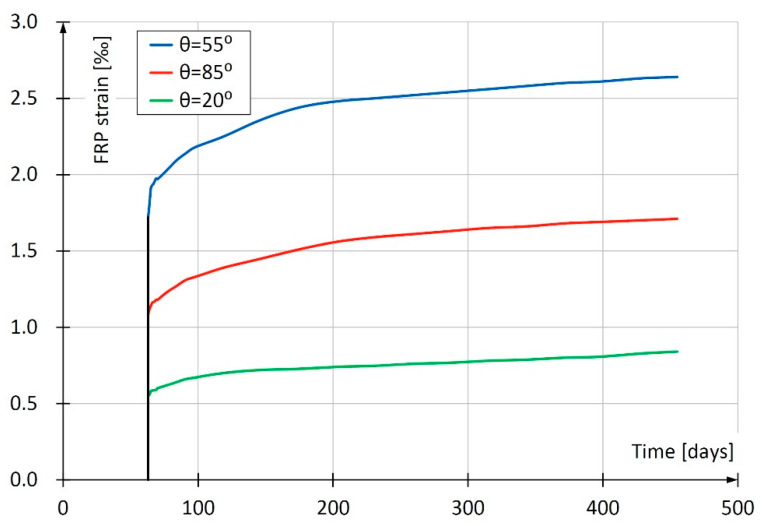
Creep curves at axial compression (t_0_ = 63 d) at stress *σ* = 0.46·*f*_c_(28).

**Table 1 materials-13-02080-t001:** Summary of the tested elements and their selected features.

No.	Element Designation	Height [m]	Angle of Cross Winding of Fibers
1	S-85	0.4	85°
2	S-55	0.4	55°
3	S-20	0.4	20°
4	L-85	2.0	85°
5	L-55	2.0	55°
6	L-20	2.0	20°

**Table 2 materials-13-02080-t002:** Compressive strength of the composite in axial compression depending on the angle of glass fiber winding.

No.	Fiber Winding Angle	Average Pipe Wall Thickness [mm]	Tube Cross-Sectional Area [mm^2^]	Destructive Force [kN]	Compressive Strength [MPa]
1	20°	7.1	4610.3	656.5	142.4
2	55°	6.5	4229.4	332.6	78.6
3	85°	5.8	3729.9	324.4	87.0

**Table 3 materials-13-02080-t003:** Computed results of longitudinal elasticity modulus and ultimate shortening of tested pipe composite during compression.

No.	Fiber Winding Angle	Elasticity Modulus	Ultimate Longitudinal Shortening
		*E*_FRP,c_ [GPa]	*ε*_FRP,u,c_ [‰]
1	20°	36.26	3.86
2	55°	14.71	7.20
3	85°	14.13	6.67

**Table 4 materials-13-02080-t004:** Medium values of mechanical parameters of pipe composite at circumferential tension stress.

Fiber Winding Angle	Modulus of Elasticity	Circumferential Stress	Ultimate Circumferential Strain
	*E* _FRP,circ_	*f* _FRP,circ_	*ε* _FRP,u,circ_
	[GPa]	[MPa]	[‰]
20°	6.02	46.1	7.66
55°	20.63	301.3	14.6
85°	46.38	692.2	14.93

**Table 5 materials-13-02080-t005:** Failure forces obtained for short concrete-filled FRP tubes (CFFT) columns depending on the angle of glass fiber winding of the composite pipe.

No.	Fiber Winding Angle	Failure Force [kN]
1	20°	1608.7
2	55°	2144.5
3	85°	4136.7

**Table 6 materials-13-02080-t006:** Failure forces obtained for long CFFT columns depending on the angle of glass fiber winding of the composite pipe.

No	Glass Fiber Winding Angle	Failure Force [kN]
1	20°	1600.0
2	55°	1406.3
3	85°	1863.7

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
