# Peer review of "Experimental Studies of Concrete-Filled Composite Tubes under Axial Short- and Long-Term Loads"

_materials, 2020, doi:10.3390/ma13092080_

Round 1

Reviewer 1 Report

The submitted manuscript aims at the experimental assessment of axially compressed columns made of concrete-filled GFRP tubes. The tested polymer pipes were reinforced with glass fiber oriented in different angles in relation to the longitudinal axis. The polymer pipes were filled with class C30/37 commercially delivered concrete. The paper is generally well written, and in the scope of the section Construction and Building Materials of Materials. The obtained results are well applicable in construction practice. The paper is well readable but the description of the conducted testes is in some cases rather confusing. For example, it is not clear, what strains were monitored in the longitudinal compression strength tests of pipes. In guess, the vertical strain is presented in Fig. 6. However, also the results of vertical strain of the tested pipes should be presented. In line 241-242, mechanical parameters of concrete used are presented. Please, add information of concrete density as usually provided in concrete testing. I don’t understand why the compressive strength and elastic modulus were tested in 86 days of concrete maturing. Please, explain. In Tables 5 and 6, failure forces are introduced. I believe, from the practical point of view, information on the strength measured for the filled pipes might be also introduced. I complete miss information on measuring uncertainty of the applied test methods. It should be provided in the description of the conducted testes and introduced also in Tables, where the measured results are presented. The legend of horizontal axis in Fig. 17 should be amended.

As the subject of the paper is of the particular interest for concrete community and the quality of the paper warrants its publication, I recommend published the paper as the above given comments will be considered in its revised version.

Reviewer 2 Report

Dear Authors,

The present paper, although it is very interesting for the scientific community, contains some shortcomings for such a publication and which need to be addressed to meet the editorial requirements of the journal.

The scientific content is very good with tables and pictures that are suggestive and cursive presented.

The abstract is too short and it could be improved.

Bibliography could be improved.

It contains only 18 references, of which 30% are standards that were mandatory.

There is too little for such a study as for the editorial requirements of the journal.

Reviewer 3 Report

This manuscript presents an experimental program on concrete-filled FRP tubes with different winding angles. The topic is important and it includes several scientific contributions. It can be accepted by addressing the following comments.

The discussions on the long-term performance should be highlighted in the conclusion part.

When carrying out the hoop behaviours of FRP tubes by tensile tests, the tested stress-strain curves should be nonlinear. In addition, filament wound FRP tube is different from other plastic material, the specimen width can certainly affect the test results. How did the author obtain the properties in Table 4? How did the authors consider these issues?

Reviewer 4 Report

the document presented is well written and detailed.
The test procedures are well explained. All the information necessary for the characterization of CFFTs is present. The composition of the concrete is given with all the parameters.
The photos and figures are of quality. you should just keep the same figure size (fig 6 and fig 17 for example). The titles of Figures 8, 11, 12, 15 seem to me too long, put the information in the text and shorten the title.
the conclusion is to review, it lacks information on the tests made on the winding angles.  

Round 2

Reviewer 3 Report

The comments by the reviewer have been addressed. There is one more strong suggestion to the authors before accepting, as listed below. The current version is more like an experimental report without deep discussion. The reviewer suggest calculating the theoretical stress-strain or load-strain curves by existing models. Then compare the theoretical curves and test results and discuss the reasons of the errors. The following two existing models are recommended for the reference. Design-oriented stress–strain model for FRP-confined concrete. Construction and building materials, 17(6-7), 471-489. Cross-sectional unification on the stress-strain model of concrete subjected to high passive confinement by fiber-reinforced polymer. Polymers, 8(5), 186.

Author Response

Dear Reviewer,

the research described in the current manuscript is indeed rather experimental.

We have already developed a theoretical model. We plan to publish this model in the near future. In our opinion it should be another article, because the authors team is different.

In addition, the research material including our theoretical model would probably be too extensive for one article.

The experimental results published in our current manuscript submitted to Materials would be a base for the future verification of our theoretical model.

We hope that our manuscript is valuable in its current form. Please consider if it is good enough for publication in Materials.

Authors

PS. We present our point of view in our separate correspondence to Academic Editor.